# Supplementation with Eupatilin during In Vitro Maturation Improves Porcine Oocyte Developmental Competence by Regulating Oxidative Stress and Endoplasmic Reticulum Stress

**DOI:** 10.3390/ani14030449

**Published:** 2024-01-30

**Authors:** Jing Wang, Ying-Hua Li, Rong-Ping Liu, Xin-Qin Wang, Mao-Bi Zhu, Xiang-Shun Cui, Zhen Dai, Nam-Hyung Kim, Yong-Nan Xu

**Affiliations:** 1Guangdong Provincial Key Laboratory of Large Animal Models for Biomedicine, School of Biotechnology and Health Sciences, Wuyi University, Jiangmen 529000, China; 2Department of Animal Science, Chungbuk National University, Cheongju 28644, Republic of Korea; 3Guangzhou Institutes of Biomedicine and Health, Chinese Academy of Sciences, Guangzhou 510530, China

**Keywords:** eupatilin, oxidative stress, in vitro maturation, oocyte

## Abstract

**Simple Summary:**

The in vitro maturation of oocytes is important for animal husbandry and assisted reproductive technology, but due to the higher oxidative stress in the in vitro environment compared with the in vivo environment, it is necessary to improve the efficiency of this process by alleviating in vitro oxidative stress. Eupatilin is a flavonoid compound with various activities, such as anti-inflammatory, anti-oxidant, and anti-tumoral activities. It is currently unclear whether it can reduce oxidative stress and improve the efficiency of the in vitro maturation of oocytes. The results of this study indicate that adding eupatilin to the in vitro maturation medium can alleviate oxidative stress, endoplasmic reticulum stress, and DNA damage in oocytes, further stimulating the maturation rate and embryo rate of porcine oocytes. It is expected to improve in vitro maturation efficiency and improve the composition of the medium.

**Abstract:**

Eupatilin (5,7-dihydroxy-3′,4′,6-trimethoxyflavone) is a flavonoid derived from *Artemisia* plants that has beneficial biological activities, such as anti-apoptotic, anti-oxidant, and anti-inflammatory activities. However, the protective effects of eupatilin against oxidative stress and endoplasmic reticulum stress in porcine oocyte maturation are still unclear. To investigate the effect of eupatilin on the development of porcine oocytes after in vitro maturation and parthenogenetic activation, we added different concentrations of eupatilin in the process of porcine oocyte maturation in vitro, and finally selected the optimal concentration following multiple comparisons and analysis of test results using SPSS (version 17.0; IBM, Chicago, IL, USA) software. The results showed that 0.1 μM eupatilin supplementation did not affect the expansion of porcine cumulus cells, but significantly increased the extrusion rate of porcine oocyte polar bodies, the subsequent blastocyst formation rate, and the quality of parthenogenetically activated porcine embryos. Additionally, it reduced the level of reactive oxygen species in cells and increased glutathione production. Further analysis revealed that eupatilin supplementation could reduce apoptosis, DNA double-strand breaks, and endoplasmic reticulum stress. In conclusion, supplementation with 0.1 μM eupatilin during in vitro maturation improved oocyte maturation and subsequent embryo development by reducing oxidative stress and endoplasmic reticulum stress.

## 1. Introduction

Oocyte maturation is an important process to ensure oocyte fertilization and early embryo development. With the development of assisted reproductive technology, scholars continue to study in vitro maturation (IVM) culture systems to improve maturation efficiency and promote in vitro embryo production [1,2]. However, the external environment surrounding the oocytes during IVM influences the normal maturation of oocytes. Previous studies have shown that oocytes are vulnerable to oxidative stress caused by reactive oxygen species (ROS). When oxidative stress is unbalanced, cells undergo apoptosis [3], mitochondrial dysfunction [4], and DNA damage [5], which hinder cell maturation. Furthermore, endoplasmic reticulum stress (ERS) in cells leads to calcium homeostasis imbalance and cell apoptosis, affecting cell maturation [6].

Eupatilin, a bioactive flavone extracted from medicinal plants belonging to the *Artemisia* genus like *Artemisia princeps* and *Artemisia copa*, has been reported to exhibit diverse beneficial biological activities encompassing neuroprotection, anti-cancerogenicity, anti-inflammatory properties, anti-oxidant functions, and free radical scavenging abilities [7,8,9,10]. Moreover, in 3T3-L1 adipocytes, eupatilin effectively reduces intracellular lipid accumulation while suppressing the expression levels of key adipogenic regulators [11]. Furthermore, eupatilin plays a key role in inhibiting the production of ROS in human bronchial epithelial cells exposed to particles and in reducing the occurrence of asthma in mice [12,13]. Additionally, eupatilin can be used as an auditory protective agent to reduce cisplatin-induced apoptosis and increase hair cell vitality [14].

While the physiological benefits of eupatilin have been extensively documented, its impact on mammalian oocytes, particularly in porcine oocytes, remains unexplored. Therefore, in order to optimize the efficiency of the IVM system, we chose to incorporate eupatilin into the IVM medium. In this study, our hypothesis was that the addition of a moderate level of eupatilin during IVM could potentially enhance the quality of porcine oocytes and promote embryonic development by alleviating oxidative stress and ERS. The objective of this study was to examine the effects of adding eupatilin during IVM on porcine oocyte development and embryonic competence after PA. This research has promising implications for improving porcine embryo production.

## 2. Materials and Methods

The chemicals and reagents used in this study were sourced from Sigma-Aldrich (St. Louis, MO, USA), unless explicitly stated.

### 2.1. Collection and IVM of Porcine Oocytes

The porcine ovaries were obtained from local slaughterhouses and promptly transported to the laboratory within 3 h in a sterile solution of 0.9% saline containing 1% penicillin–streptomycin at a controlled temperature of 30–35 °C. The follicles, ranging from 3 to 6 mm in diameter, were aspirated using a 10 mL syringe equipped with an 18-gauge needle. The IVM of porcine oocytes was conducted following a previously documented protocol [15]. Briefly, cumulus–oocyte complexes (COCs) were collected under a stereomicroscope if they had three or more uniformly distributed layers of cumulus cells, and were then subjected to 44 h of in vitro cultivation.

The degree of expansion in cumulus cells was assessed at the conclusion of IVM using a microscope (Eclipse Ti2; Nikon, Tokyo, Japan). The degree of cumulus expansion was assessed by utilizing ImageJ software (NIH, Bethesda, MD, USA, https://ij.imjoy.io/) to measure the sizes of oocytes and their corresponding cumulus cells. Subsequently, COCs with an enlarged cumulus–corona cell complex were disintegrated using a 0.1% hyaluronidase solution, followed by the examination of polar body extrusion (PBE) in oocytes using a stereomicroscope (SMZ745; Nikon, Tokyo, Japan).

### 2.2. Eupatilin Treatment

The compound eupatilin (#HY-N0783, MedChemexpress, Monmouth Junction, NJ, USA) was solubilized in DMSO and subsequently diluted in IVM medium to achieve the final concentrations of 0.01, 0.1, and 1 μM. The COCs that were gathered underwent incubation in a preheated IVM solution and were cultured within an incubator set at 38.5 °C, with a CO_2_ concentration of 5%. Control oocytes matured without the addition of eupatilin were included for comparison.

### 2.3. PA of Oocytes and In Vitro Culture (IVC)

Following the removal of cumulus cells, as previously described in our research, the detached oocytes underwent a PA procedure [15]. In brief, oocytes were activated by the application of two direct current pulses at a voltage of 120 V for a duration of 60 μs. The purportedly activated oocytes were cultured in porcine zygote medium-5 (PZM-5) supplemented with 4 mg/mL bovine serum albumin (BSA) and 7.5 μg/mL cytochalasin B for a duration of three hours to hinder the extrusion of the pseudo-second polar body. Subsequently, the oocytes were thoroughly washed and transferred to a four-well plate containing PZM-5 medium covered with mineral oil for further cultivation under controlled conditions of 38.5 °C, 100% humidity, and a CO_2_ atmosphere maintained at a concentration of 5%, without altering the culture medium. The rates of cleavage and blastocyst formation on days 2 and 7 were evaluated using stereomicroscopy.

### 2.4. Evaluation of Blastocyst Dimensions and Total Cell Counts

On day 7, blastocysts were imaged using a microscope, and the blastocyst diameter in each group was quantified using ImageJ software. To determine the total cell count within the blastocysts, parthenogenetically activated embryos’ day 7 blastocysts were collected and fixed in a medium containing 0.1% polyvinyl alcohol in phosphate-buffered saline (PBS-PVA) with 3.7% paraformaldehyde (FA) for 30 min. Subsequently, they were incubated at room temperature for 30 min in a solution of 0.3% Triton X-100. Following this incubation period, the blastocysts were stained with Hoechst 33342 at a concentration of 10 μg/mL and further incubated at 37 °C under dark conditions for 15 min. The stained blastocysts were gently mounted on slides and examined as well as photographed using fluorescence microscopy. The total cell count within the blastocysts was determined by employing ImageJ software.

### 2.5. TUNEL Assay

As per the guidelines provided by the manufacturer, we employed a TUNEL assay kit (#11684795910; Roche, Basel, Switzerland) to examine cellular apoptosis. The day 7 blastocysts were subjected to fixation in a solution of 3.7% formaldehyde at room temperature for a duration of 30 min. This was followed by an additional incubation period of 30 min at room temperature in a solution containing 0.3% Triton X-100. Afterwards, the blastocysts were placed in a dark environment at a temperature of 37 °C for one hour, where they were exposed to fluorescein-conjugated dUTP and terminal deoxynucleotidyl transferase enzyme. Subsequently, the cells underwent three rounds of washing using a 0.1% PBS-PVA solution. The Hoechst 33342 was applied to the embryos at a concentration of 10 μg/mL, followed by incubation in darkness for 15 min at a temperature of 37 °C. Then, the cells were rinsed thrice using 0.1% PBS-PVA solution and subsequently placed on a slide for sealing. The fluorescence intensity, apoptotic nuclei count, and total observed nuclei were quantitatively assessed using fluorescent microscopy and ImageJ software. The apoptosis evaluation was based on the proportion of apoptotic nuclei within blastocysts.

### 2.6. Intracellular ROS and Glutathione (GSH) Level Assays

At the conclusion of the IVM phase, in accordance with the instructions provided by the reagent kit, the intracellular ROS and GSH levels were quantified using Thermo Fisher Scientific (Waltham, MA, USA) (# C400) and GSH detection kits (Thermo Fisher Scientific; # C12881). A digital camera connected to a fluorescence microscope was employed to capture TIFF images of the fluorescence signal. Subsequently, NIH ImageJ software was utilized for analysis of fluorescence signal intensity within each oocyte group.

### 2.7. Measurement of Cathepsin B (CB) Activity

The measurement of CB activity was conducted employing a CB assay kit (Immuno Chemistry, Davis, CA, USA; #938) following the guidelines provided by the manufacturer. After a 44 h culture period, oocytes without their outer layer were placed in a solution of 0.1% PBS-PVA and incubated with a reaction mix for 30 min at 37 °C in the absence of light. The fluorescence signals were captured using fluorescence microscopy and subsequently analyzed using ImageJ software.

### 2.8. Immunofluorescence Staining

After maturation of the oocytes, cumulus cells were eliminated using hyaluronidase. After being rinsed three times with 0.1% PBS-PVA solution, the oocytes were fixed in 3.7% FA for a duration of 30 min at room temperature. Subsequently, they were made permeable by treating them with a solution containing 0.1% PBS-PVA and 0.3% Triton X-100 for half an hour at room temperature. Following this step, the oocytes were incubated in a solution consisting of 1% BSA in PBS-PVA for one hour at a temperature of 37 °C. The oocytes were subjected to overnight incubation at 4 °C with a mouse monoclonal antibody against γH2A.X (Abcam, Cambridge, UK; #ab26350; diluted 1:100). This was followed by subsequent incubation with a secondary antibody conjugated to Alexa Fluor 594 (Cell Signaling Technology, Danvers, MA, USA; #8890; diluted 1:1000) for an hour at 25 °C. The oocytes were subsequently treated with Hoechst 33342 at a concentration of 10 μg/mL, and incubated in the dark at 37 °C for 15 min. Following this, the cells underwent three washes with a solution of 0.1% PBS-PVA before being mounted on a slide for sealing; thereafter, they were imaged under fluorescent light using a microscope. The γH2A.X-negative/-positive signal was determined based on the analysis of fluorescence intensity.

### 2.9. RNA Extraction and qRT-PCR Assay

The Dynabeads mRNA DIRECT purification kit (Invitrogen, Carlsbad CA, USA; #61012) was utilized to extract the overall mRNA from around 200 MII oocytes. Subsequently, the RNA that was obtained was converted into cDNA through reverse transcription using the Dynabeads mRNA direct kit (Invitrogen; #18080-051). The qPCR experiment was conducted using a CFX Connect Optics Module (Roche/Light Cycler 96) and a Kapa kit (South Africa; Swiss Confederation; #KK4600). The qRT-PCR reaction mixture (20 µL) was composed of 8 µL of deionized water, 10 µL of SYBR Green, 1 µL of cDNA, and 0.5 µL each of the forward and reverse primers (10 mM). The thermal cycler was programmed to initiate denaturation at a temperature of 95 °C for a duration of 3 min. This was followed by a series of 40 cycles consisting of denaturation at the same temperature for a brief period of 3 s, annealing at 60 °C for approximately half a minute, and extension at 72 °C for about 20 s. The target genes included those associated with the process of oocyte maturation. The target genes comprised oocyte maturation-associated genes (proto-oncogene serine/threonine kinase (*MOS*), cyclinB1 (*CCNB1*), and bone morphogenetic protein 15 (*BMP15*)), oxidative stress-related genes (catalase (*CAT*), sirtuin 1 (*SIRT1*), and superoxide dismutase 1 (*SOD1*)), ERS-related genes (glucose-regulated protein 78 (*GRP78*), inositol-requiring enzyme 1 (*IRE1*), activating transcription factor 6 (*ATF6*), and c-Jun N-terminal kinase (*JNK*)), apoptosis-related genes (Bcl-2-associated X (*BAX*) and *Caspase-3*), and blastocyst pluripotency-related genes (*NANOG* and octamer-binding transcription factor 4 (*OCT4*)). The glyceraldehyde-3-phosphate dehydrogenase gene (*GAPDH*) was utilized as a reference gene. The primers employed for amplification of each gene are presented in Table 1. mRNA quantification data were analyzed using the 2^−ΔΔCT^ method.

### 2.10. Statistical Analysis

The results are reported as means ± standard deviations (SDs). The figures illustrate the total number of MⅡ oocytes and blastocysts utilized (n), as well as the number of independent repetitions (R) conducted for each experiment. The statistical analysis employed in this study involved conducting a one-way analysis of variance, followed by Tukey–Kramer tests, to evaluate significant differences and compare individual means when appropriate. The statistical analyses were performed using SPSS. The significance levels are indicated as follows: * *p* < 0.05; ** *p* < 0.01; and *** *p* < 0.001.

## 3. Results

### 3.1. Effects of Eupatilin Supplementation on Porcine Oocyte Maturation during IVM

To investigate the impact of eupatilin supplementation on porcine oocyte quality during IVM, COCs were cultured with varying concentrations of eupatilin (0.01, 0.1, and 1 μM). At the end of IVM, evaluations were conducted on the cumulus expansion status, PBE rate, and gene expression in porcine oocytes. Representative images illustrating cumulus expansion status are presented in Figure 1A. No significant differences were observed in the relative cumulus cell expansion areas between the control group and groups supplemented with eupatilin (Figure 1B; *p* > 0.05). However, 0.1 μM eupatilin supplementation significantly increased the PBE rate of oocytes (Figure 1C; control, 69.12 ± 6.11%; 0.01 μM, 70.23 ± 5.06%; 0.1 μM, 82.51 ± 4.04% vs. 1 μM, 60.86 ± 7.96%; *p* < 0.001). In addition, the mRNA expression levels of the *MOS*, *CCNB1*, and *BMP15* genes were significantly upregulated in the 0.1 μM eupatilin-supplemented group compared with the control group (Figure 1D; control, 1.00 ± 0.00; *MOS*, 1.21 ± 0.05; *CCNB1*, 1.35 ± 0.08; *BMP15*, 1.22 ± 0.01; for all, *p* < 0.05).

### 3.2. Effects of Eupatilin Supplementation during IVM on Porcine Embryonic Developmental Capacity

We explored whether the addition of eupatilin during IVM could improve the developmental ability of porcine embryos. The blastocyst rate of parthenogenetic embryos produced by supplementing mature oocytes with 0.1 μM eupatilin was significantly higher than that of the control group (Figure 2A,C; 44.92 ± 4.01% vs. 32.60 ± 4.28% on day 7; *p* < 0.001), but supplementation had no significant effects on the cleavage rate (Figure 2B, *p* > 0.05). Further analysis showed that treatment with 0.1 μM eupatilin during IVM sharply increased blastocyst diameter (Figure 2D; 208.45 ± 42.82 vs. 192.31 ± 42.89 μm on day 7; *p* < 0.05) and blastocyst cell number (Figure 3B; 48.31 ± 10.11 vs. 39.09 ± 6.35 on day 7; *p* < 0.001). Additionally, it reduced apoptosis and the expression level of apoptotic genes (*BAX* and *Caspase-3*) in blastocyst cells and promoted the expression of pluripotency-related genes (*NANOG* and *OCT4*; Figure 3A,C and Figure 4; *p* < 0.01). Given these findings, 0.1 μM eupatilin was selected for use in all subsequent experiments.

### 3.3. Effects of Eupatilin Supplementation during IVM on Oxidative Stress Resistance in Porcine Oocytes

Oxidative stress in oocytes during IVM is considered to be the key factor affecting oocyte quality. To examine whether eupatilin could reduce oxidative stress during IVM, H_2_DCFDA and CMF_2_HC fluorescent probes were used to detect the ROS and GSH levels in oocytes. The relative fluorescence levels of H_2_DCFDA and CMF_2_HC in eupatilin-treated oocytes were significantly decreased and increased, respectively, compared to those in the control group (0.73 ± 0.22 vs. 1.00 ± 0.23 (Figure 5A,C); 1.63 ± 0.31 vs. 1.00 ± 0.15 (Figure 5B,D); both *p* < 0.001). The molecular analysis of the anti-oxidant enzymes revealed that the expression levels of catalase (*CAT*; 1.00 ± 0.00 vs. 1.24 ± 0.09), silent information regulator 1 (*SIRT1*; 1.00 ± 0.00 vs. 1.50 ± 0.08), and superoxide dismutase 1 (*SOD1*; 1.00 ± 0.00 vs. 1.43 ± 0.11) in oocytes within the eupatilin-supplemented group exhibited a significant increase compared to those in the control group (Figure 5E; *p* < 0.05; *p* < 0.01).

### 3.4. Effects of Eupatilin Supplementation during IVM on Apoptosis of Porcine Oocytes

The occurrence of oxidative stress triggers the process of apoptosis. The CB activity in the group supplemented with eupatilin exhibited a significant decrease compared to that in the control group (Figure 6A,B; 0.64 ± 0.16 vs. 1.00 ± 0.12; *p* < 0.001). Additionally, we conducted RT-qPCR to assess the mRNA expression levels of the pro-apoptotic genes, *Caspase-3* and *BAX*. The findings demonstrate that supplementation of eupatilin resulted in a reduction in the expression of pro-apoptotic genes and an increase in the expression of anti-apoptotic genes (Figure 6C; *p* < 0.05).

### 3.5. Effects of Eupatilin Supplementation during IVM on DNA Double-Strand Breaks in Porcine Oocytes

DNA double-strand breaks affect oocyte maturation. Therefore, we measured the expression of DNA double-strand break marker γH2A.X in the different groups. The results show that 26.5 ± 1.41% and 19.93 ± 3.20% of oocytes were positive in the control group and eupatilin group, respectively (Figure 7A,B; *p* < 0.05).

### 3.6. Effects of Eupatilin Supplementation during IVM on ERS-Related Genes in Porcine Oocytes

In order to detect the effects of eupatilin supplementation during IVM on ERS in porcine oocytes, we measured the expression of ERS-related genes, namely, *GRP78*, *IREI*, *ATF6*, and *JNK*. We observed that eupatilin supplementation significantly decreased the expression of ERS-related genes (Figure 8; *p* < 0.05).

## 4. Discussion

The IVM of oocytes is the most critical step of in vitro embryo production. It involves the maturation of the oocyte nucleus and cytoplasm. Due to interference from the external environment, maturation efficiency is reduced. Therefore, we chose to add a Chinese herbal extract to the IVM culture solution to explore its impact on oocyte maturation and subsequent embryonic development. The results showed that eupatilin reduced oxidative stress and ERS, reduced apoptosis, promoted embryonic development, and improved oocyte quality (Figure 9).

During IVC, uncertain factors interfere with oocyte maturation. Previous studies have demonstrated that adding anti-oxidants during IVM can improve embryo development [16,17,18]. In this study, we analyzed the extent of cumulus expansion, oocyte PBE rate, and maturation-promoting factor (MPF) activity at the end of IVM. The addition of 0.1 μM eupatilin significantly increased the PBE rate and the expression of MPF activity (*MOS*, *CCNB1*, *BMP15*), but had no significant effects on cumulus expansion. In mammalian species, the activation of *MOS* expression occurs promptly after the breakdown of oocyte germinal vesicles, and is essential for facilitating oocyte maturation through its involvement in the activation and/or stabilization of MPF. [19]. The *BMP15* protein, belonging to the transforming growth factor-beta family, plays a crucial role in promoting oocyte maturation and cumulus expansion across various species including pigs, dogs, mice, sheep, and humans. [20,21]. Moreover, stable *MOS* and *CCNB1* levels are essential for meiosis mediated by the spindle checkpoint protein MAD1, monopolar spindle 1 kinase, and APC/CCdc20 Moreover, maintaining stable levels of *MOS* and *CCNB1* is crucial for meiosis regulated by the spindle checkpoint protein MAD1, monopolar spindle 1 kinase, and APC/CCdc20. [22]. Further analysis showed that the blastocyst formation rate, diameter, and cell number were significantly enhanced, and that the apoptosis rate was significantly decreased when eupatilin was supplemented during IVM after PA. Additionally, it reduced the expression of apoptotic genes and increased the expression of pluripotent genes. The findings of this study provide support for our hypothesis that the addition of eupatilin during IVM enhances the quality of mature porcine oocytes, and improves their ability to develop into embryos.

Porcine oocytes have more lipid droplets and undergo a longer maturation period than the oocytes of other mammals [23,24], which is why they are more vulnerable to oxidative stress. When the level of ROS in cells is imbalanced, it leads to apoptosis, mitochondrial dysfunction, and meiotic process disruption, which, in turn, affect embryonic development [3,25,26,27]. GSH is a major cellular redox regulator that controls redox balance, and it is one of the key indicators of cytoplasmic maturation in oocytes [28]. Our investigation of intracellular ROS and GSH levels showed that supplementation with eupatilin during IVM improved the developmental capacity of porcine oocytes, and may have been responsible for the improvement in oxidative stress in the eupatilin supplementation group.

The results of our study demonstrate that the addition of eupatilin led to a reduction in the expression levels of oxidative stress-related genes (*CAT*, *SIRT1*, *SOD1*), and an increase in the expression levels of anti-oxidant stress genes. Recent studies have demonstrated that supplementation of eupatilin significantly attenuates the production of nitric oxide, interleukin-6, and reactive oxygen species in lipopolysaccharide-stimulated RAW 264.7 cells [13]. Previous research has also demonstrated a dose-dependent decrease in the production of ROS in a human bronchial epithelial cell line (BEAS-2B) when exposed to fine particulate matter, which may contribute to the alleviation of respiratory diseases [12]. Furthermore, eupatilin exerted a protective effect against hydrogen peroxide (H2O2)-induced oxidative stress and apoptosis by activating the PI3K/Akt signaling pathway in ARPE-19, a human retinal pigment epithelium cell line [29]. The above results are consistent with our own. Supplementing the IVM medium with eupatilin can improve oxidative stress and oocyte quality.

The endoplasmic reticulum (ER) plays a pivotal role in oocyte meiotic maturation by balancing the cytoplasmic Ca^2+^ concentration and regulating protein synthesis, folding, and maturation [30,31]. It was reported that severe hypoxia, oxidative injury, or cytotoxins impaired ER homeostasis by activating the ERS-mediated unfolded protein response (UPR) during reproduction [32]. *PERK*, *IER1*, and *ATF6* located on the ER jointly regulate the UPR signal, but when the UPR is above a certain threshold, unresolved ERS may further lead to apoptosis and DNA damage [33,34]. In the current study, we found that eupatilin supplementation can reduce the expression of ERS genes, cell apoptosis, and DNA damage. This suggests that eupatilin protects the homeostasis of the ER, and further protects oocyte maturation.

## 5. Conclusions

In conclusion, the present results show that eupatilin supplementation in the IVM medium benefits the maturation of porcine oocytes. In particular, it can prevent apoptosis and DNA damage mediated by oxidative stress and ERS, and can promote oocyte maturation and embryonic development.

## Figures and Tables

**Figure 1 animals-14-00449-f001:**
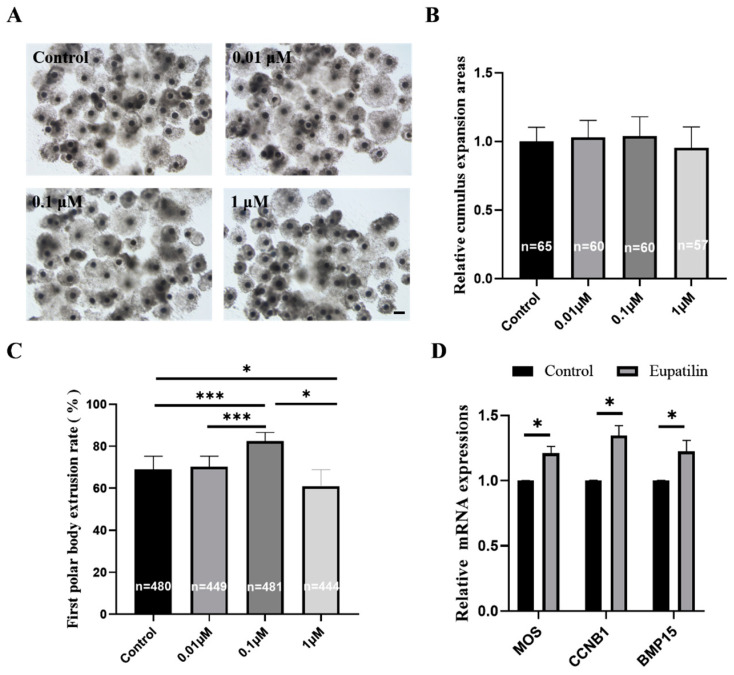
Effects of eupatilin supplementation during IVM on cumulus cell expansion status, oocyte maturation, and gene expression. (**A**) Representative images depicting the expansion of cumulus cells in control and eupatilin-supplemented oocytes. Scale bar = 200 μm. (**B**) Relative expansion areas of cumulus cells in the control and eupatilin groups were compared (R = 3). (**C**) The impact of varying concentrations of eupatilin on the maturation rate of porcine oocytes (R = 8; * *p* < 0.05; *** *p* < 0.001). (**D**) Relevance of oocyte competence-related genes following supplementation with eupatilin (R = 3; * *p* < 0.05). The number of oocytes examined in the different groups is indicated on the bars in (**B**,**C**).

**Figure 2 animals-14-00449-f002:**
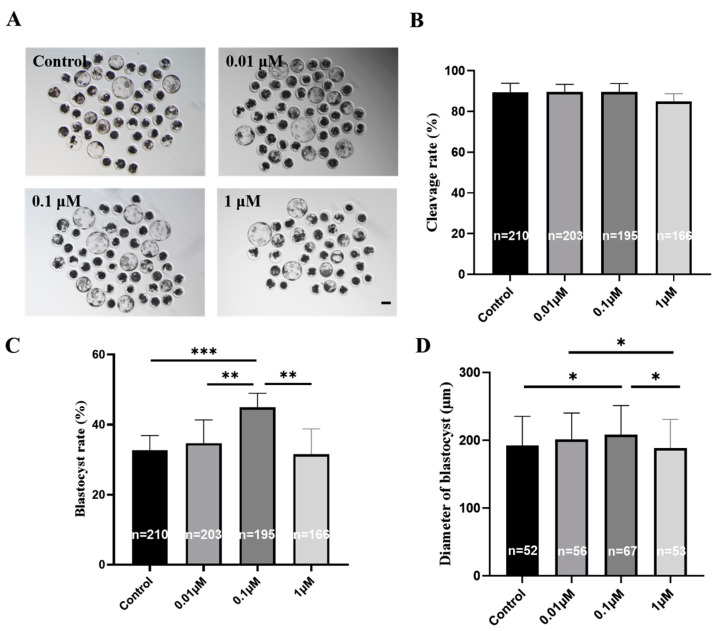
Impact of eupatilin supplementation during IVM on the developmental competence of porcine preimplantation embryos. (**A**) Relevant images depicting the developmental progression of embryos on day 7 in both the control and eupatilin-supplemented groups were obtained (0.01 μM, 0.1 μM, 1 μM). Scale bar = 100 μm. (**B**) Reproductive division rates in the control and eupatilin groups (R = 6). (**C**) Blastocyst rates in the control and eupatilin groups were assessed (R = 6; ** *p* < 0.01, *** *p* < 0.001). (**D**) Diameters of blastocysts in the control and eupatilin groups were measured (R = 5; * *p* < 0.05). The numbers of oocytes analyzed in the various groups are specified on the bars in (**B**–**D**).

**Figure 3 animals-14-00449-f003:**
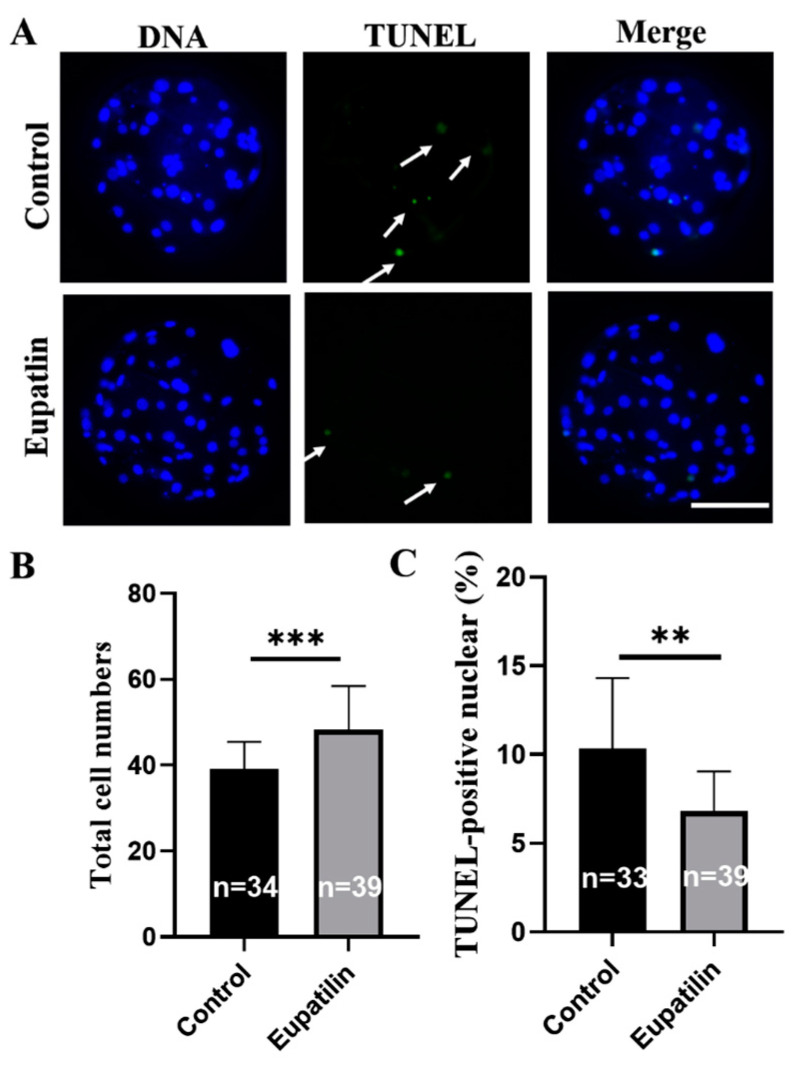
Impact of eupatilin supplementation during IVM on the total cell numbers and apoptotic nuclei of porcine PA embryos. (**A**) Relevant images depicting the TUNEL staining of blastocysts on day 7 in both the control and eupatilin groups were obtained for analysis. The white arrows indicate nuclei positively stained for apoptosis. Scale bar = 100 μm. (**B**) Total cell numbers of blastocysts in the control and eupatilin groups were assessed (R = 4; *** *p* < 0.001). (**C**) The proportions of apoptotic cells in blastocysts were compared between the control group and the eupatilin-treated group (R = 4; ** *p* < 0.01). The number of oocytes examined in each group is indicated on the corresponding bars in (**B**,**C**).

**Figure 4 animals-14-00449-f004:**
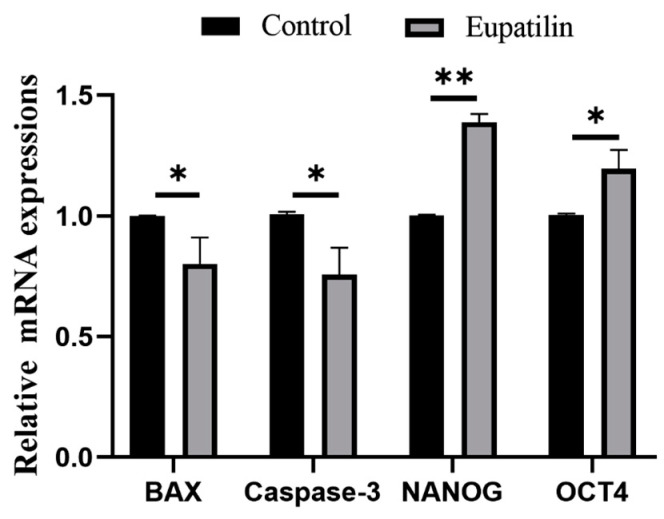
Effects of eupatilin supplementation during IVM on mRNA levels of pluripotency- and apoptosis-related genes in porcine PA embryos (R = 3; * *p* < 0.05; ** *p* < 0.01).

**Figure 5 animals-14-00449-f005:**
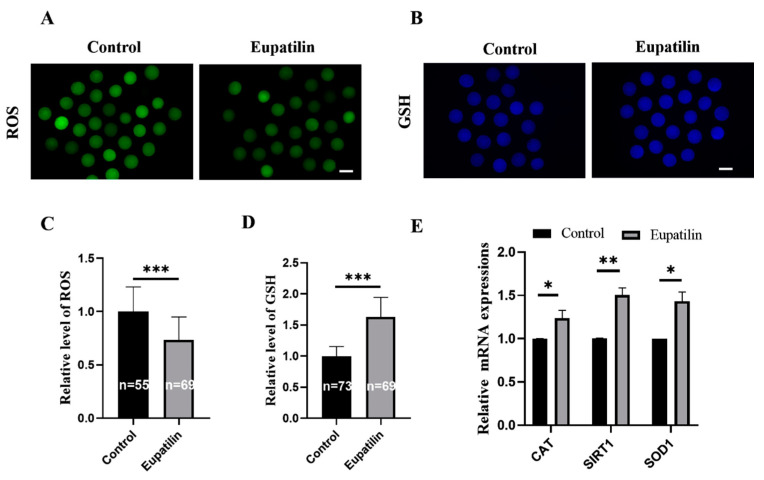
Impact of eupatilin supplementation on intracellular ROS levels, GSH activity, and gene expression in porcine oocytes. (**A**,**B**) Representative fluorescence images depicting intracellular levels of ROS and GSH-stained porcine oocytes were captured in both the control group and the eupatilin-supplemented group at the conclusion of IVM. Scale bar = 100 μm. (**C**,**D**) Quantification of relative intracellular levels of ROS and reduced GSH in porcine oocytes in both the control group and the eupatilin-supplemented group. (**E**) mRNA expression levels of genes associated with oxidative stress. (R = 3; * *p* < 0.05; ** *p* < 0.01; *** *p* < 0.001). The number of oocytes examined in each group is indicated on the corresponding bars in (**C**,**D**).

**Figure 6 animals-14-00449-f006:**
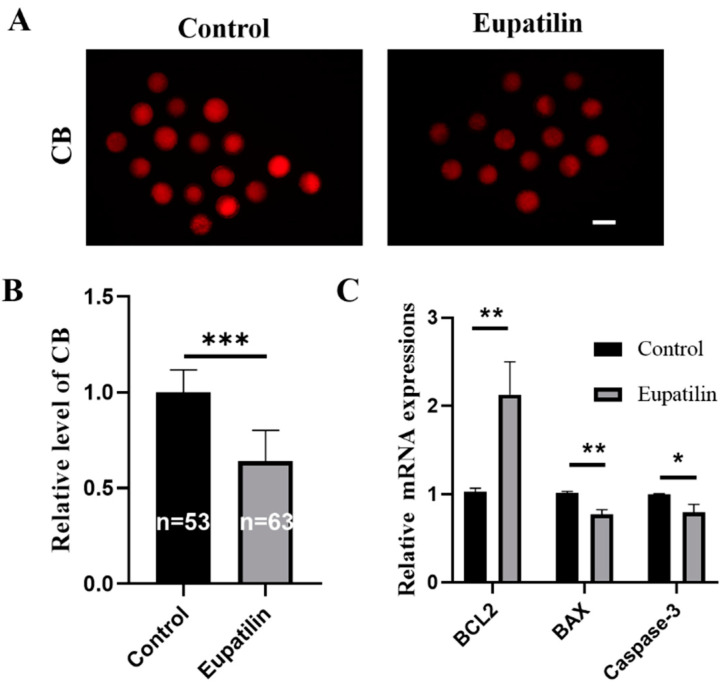
Effects of eupatilin supplementation on intracellular CB activity and the expression of associated genes. (**A**) Representative fluorescence images depicting the CB activity of porcine oocytes in both the control and eupatilin-supplemented groups were captured. Scale bar = 100 μm. (**B**) Quantification of relative intracellular CB levels in porcine oocytes was performed in the control group and groups supplemented with eupatilin (R = 3; *** *p* < 0.001). The number of oocytes examined in each group is indicated on the corresponding bars. (**C**) Expression levels of genes associated with apoptosis at the mRNA level (R = 3; * *p* < 0.05, ** *p* < 0.01).

**Figure 7 animals-14-00449-f007:**
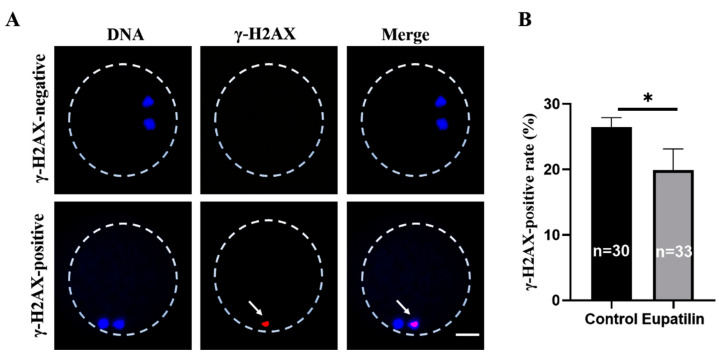
Effects of eupatilin supplementation on DNA double-strand breaks in porcine oocytes. (**A**) γH2A.X-negative/-positive representative fluorescent images of porcine oocytes in control and eupatilin-supplemented groups. Scale bar = 25 μm. The white arrows indicate nuclei positively stained for γH2A.X. (**B**) Ratio of γH2A.X focus-positive oocytes in the control and eupatilin-supplemented groups. The number of oocytes examined in each group is indicated on the corresponding bars (R = 3; * *p* < 0.05).

**Figure 8 animals-14-00449-f008:**
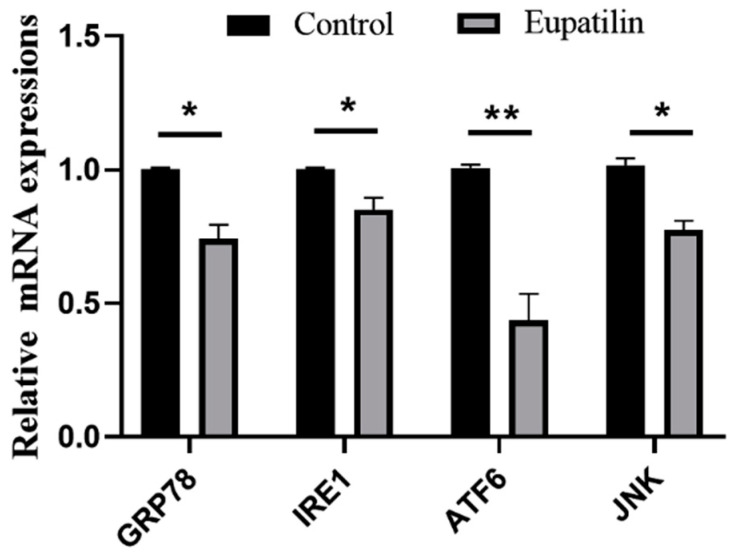
mRNA levels of ERS-related genes (R = 3; * *p* < 0.05; ** *p* < 0.01).

**Figure 9 animals-14-00449-f009:**
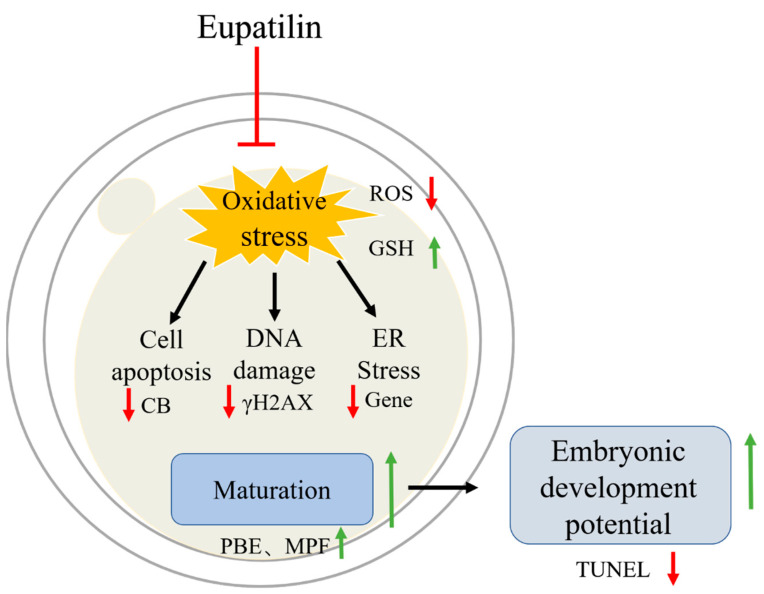
Hypothetical model of the anti-oxidant effect of eupatilin on porcine oocyte maturation. Eupatilin can improve the quality of porcine oocytes and embryonic developmental ability by reducing oxidative damage. Eupatilin prevents oxidative stress, avoids apoptosis, DNA damage, and ERS, and improves oocyte maturation and embryonic development.

**Table 1 animals-14-00449-t001:** Primers for qRT-PCR.

Gene	Sequences 5′–3′	Product Size (bp)	Accession Number
*GAPDH*	F: TTCCACGGCACAGTCAAG	117	NM_001206359.1
	R: ATACTCAGCACCAGCATCG		
*MOS*	F: GGTGGTGGCCTACAATCTCC	136	NM_001113219.1
	R: TCAGCTTGTAGAGCGCGAAG		
*CCNB1*	F: CCAACTGGTTGGTGTCACTG	195	NM_001170768.1
	R: GCTCTCCGAAGAAAATGCAG		
*BMP15*	F: ATGCTGGAGTTGTACCAGCG	87	NM_001005155.2
	R: CTGAGAGGCCTTGCTCCATT		
*CAT*	F: AACTGTCCCTTCCGTGCTA R: CCTGGGTGACATTATCTTCG	83	XM_021081498.1
*SIRT1*	F: GAGAAGGAAACAATGGGCCG R: ACCAAACAGAAGGTTATCTCGGT	150	NM_001145750.2
*SOD1*	F: CAAAGGATCAAGAGAGGCACG R: CGAGAGGGCGATCACAGAAT	84	NM_001190422.1
*BAX*	F: GCTTCAGGGTTTCATCCAGGATCG R: ACTCGCTCAACTTCTTGGTAGATC	107	XM_003127290.5
*Caspase3*	F: TGTGGGATTGAGACGGACAG R: TTTCGCCAGGAATAGTAACCAGG	116	NM_214131.1
*GRP78*	F: CGGAGGAGGAGGACAAGAAGGAG R: ATATGACGGCGTGATGCGGTTG	143	XM_001927795.7
*IRE1*	F: ACCGTGGTGTCTCAGGATGTGG R: CCAGCCAATGAGCAGGAAGGTG	126	XM_005668695.3
*JNK*	F: CTCGCTACTACAGAGCACCTG R: TTCTCCCATAATGCACCCCAC	85	XM_021073087.1
*ATF6*	F: GGAGTTAAGACAGCGCTTGG R: GAGATGTTCTGGAGGGGTGA	142	NM_001271738.1
*NANOG*	F: TGTCTCTCCTCTTCCTTCCTCCATG R: TCCTCCTTCTCTGTGCTCTTCTCTG	117	NM_001129971.1
*OCT4*	F: CCTATGACTTCTGCGGAGGGA R: TTTGATGTCCTGGGACTCCTCG	224	XM_021097869.1

F: forward primer; R: reverse primer. The annealing temperature for all reactions was 60 °C.

## Data Availability

The data presented in this study are available upon request from the corresponding author. The data are not publicly available due to the confidentiality of the manuscript.

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
