# Peer review of "Supplementation with Eupatilin during In Vitro Maturation Improves Porcine Oocyte Developmental Competence by Regulating Oxidative Stress and Endoplasmic Reticulum Stress"

_animals, 2024, doi:10.3390/ani14030449_

Round 1

Reviewer 1 Report

Comments and Suggestions for Authors

Comments on the Quality of English Language

Author Response

Response to Reviewer 1 Comments

The manuscript describes the examination of the impact of eupatilin on porcine oocyte development during in vitro maturation. Eupatilin is a flavone from Artemisia asiatica and has potential medical properties, including reducing oxidative stress. Studies included determining eupatilin effect on the cumulus expansion status, polar body extrusion, parthenogenetic activation, oxidative stress resistance and apoptosis process in the porcine oocytes. The obtained results may have an impact on the improvement of in vitro oocyte maturation.

Overall, the article is interesting but some points should be completed before publication.

Respond:We appreciate the reviewer’s comments. We have revised the manuscript based on your suggestion.

Point 1:

  • The manuscript would benefit from English editing by a native speaker.

Respond 1:We appreciate the reviewer’s comments. We use the English editing website recommended by MDPI to edit articles in English. Thank you for your suggestion to make it better published in Animals magazine.

Point 2:

  • The first sentence in Summary is incomprehensible. Shouldn't it be: "in the in vitro environment compared to in vivo…" instead of " in the in vitro environment compared to in vitro…"?

Respond 2:We appreciate the reviewer’s comments. We have revised the sentence based on your suggestion.

Point 3:

  • It seems to me that it would be advantageous to add to the introduction why these specific genes (MOS, CCNB1, BMP15, CAT, SIRT1, SOD1, BAX, Caspase3, GPR78, IRE1, JNK, ATF6, NANOG, OCT4) were selected in the experiment.

Respond 3:We appreciate the reviewer’s comments. But we provided a brief explanation of these genes in lines 161-164 on page four, as we conducted real-time fluorescence quantitative PCR on multiple genes during the experiment, so we did not introduce the genes in the introduction section. Thank you very much for your suggestion.

Point 4:

  • The article should be corrected in terms of introduced and explained abbreviations. Once introduced, an abbreviation should be used consistently later in the manuscript. Furthermore, I think that it would be better to explain the abbreviations used in figures and tables in their respective descriptions.

Respond 4:We appreciate the reviewer’s comments. We have revised the manuscript based on your suggestion.

Point 5:

  • Please correct the description by indicating the size of the mature follicles from which COCs were collected and the size of the complexes that were collected.

Respond 5:We appreciate the reviewer’s comments. We have made modifications based on your suggestions.

Page 2, lines 74-78:

Follicles with a diameter of 3 to 6 mm were aspirated with a 10 mL syringe with an 18-gauge needle.

Briefly, we collected cumulus–oocyte complexes (COCs) with three or more uniformly distributed layers of cumulus cells under a stereomicroscope for 44-hour in vitro cultivation.

Point 6:

  • The information provided in lines 171-173 about concentrations of eupatilin used in the experiment should be included in the IVM description in the materials and methods section.Why were these concentrations chosen? Where was the eupatilin obtained, was it commercially purchased or self-obtained?

Respond 6:We appreciate the reviewer’s comments. We have supplemented the information on eupatilin based on your suggestion.

Page 2, lines 86-90:

Eupatilin(#HY-N0783,MedChemexpress,USA) was dissolved in DMSO and then diluted in IVM medium to produce the final concentrations of 0.01, 0.1, and 1 μM. The collected COCs were incubated in preheated IVM medium and cultured in a 38.5 ℃, 5% CO2 incubator. Oocytes matured without eupatilin supplementation served as controls.

During the experiment, the concentration of eupatilin was selected from 0, 50, and 100 μM, and the concentration range was continuously reduced. Finally, based on the polar body excretion rate and blastocyst rate, select 0.1 μM as the best supplement. Eupatilin was purchased from MedChemExpress company.

Point 7:

  • Lines 91-93: This sentence is unclear. Were oocytes washed and cultured without changing the medium? Did the Authors mean: without changing the composition of the medium?

Respond 7:We appreciate the reviewer’s comments. Yes, oocytes activated by parthenogenesis were cultured for 7 days without changing the medium.

Point 8:

  • The first sentence in the paragraph "Assessment of blastocyst diameters and total cell numbers" in the materials and methods section (lines 96-97) is redundant because this information is repeated later. It should be removed from the manuscript.

Respond 8:We appreciate the reviewer’s comments. We have made the deletion according to your suggestion.

Point 9:

  • Lines 109-110: This sentence is unclear. Should not the samples along with fluorescein coupled dUTP be incubated?

Respond 9:We appreciate the reviewer’s comments. Thank you very much for making the modifications.

Page 3, lines 116-118:

Then, the blastocysts were cultured for 1 h at 37 °C in the dark while being exposed to fluorescein-conjugated dUTP and terminal deoxynucleotidyl transferase enzyme.

Point 10:

  • Why did the incubation with secondary antibodies in immunofluorescence analysis take 1-2 hours, and what did it depend on? All settings compared in the experiment should be performed under identical conditions.

Respond 10:We appreciate the reviewer’s comments. The experiment was conducted according to the suggestions in the instruction manual. The following experiments were conducted after 1 hour of incubation. The content of the article has been revised. Thank you for the reviewer's suggestions.

Point 11:

  • Line 183: The values described for CCNB1 (1.35±0.76) do not correspond to the values observed in the diagram (Fig.1D).

Respond 11:We appreciate the reviewer’s comments. We have reanalyzed the data and have corrected the correct data. Thank you to the reviewer for your valuable suggestions.

Point 12:

  • Why was real-time PCR analysis performed only with R=3 and only with one reference gene? The real-time PCR method should meet MIQE guidelines. Additionally, why is there a different R in each experiment?

Respond 12:We appreciate the reviewer’s comments. During the experiment, each qRT PCR was repeated three times, and the average value was taken as the data. A total of three repetitions were used as the final experimental result.

Most articles use an internal reference gene, so this experiment also used an internal reference gene[1-3].

Because in the early stage of screening for the concentration of eupatilin, it is necessary to repeat it several times to avoid errors.

References:

  1. Wang, Y.; Qi, J.J.; Yin, Y.J.; Jiang, H.; Zhang, J.B.; Liang, S.; Yuan, B. Ferulic Acid Enhances Oocyte Maturation and the Subsequent Development of Bovine Oocytes. International journal of molecular sciences 2023, 24, doi:10.3390/ijms241914804.
  2. Li, Z.; Chen, C.; Yu, W.; Xu, L.; Jia, H.; Wang, C.; Pei, N.; Liu, Z.; Luo, D.; Wang, J.; et al. Colitis-Mediated Dysbiosis of the Intestinal Flora and Impaired Vitamin A Absorption Reduce Ovarian Function in Mice. Nutrients 2023, 15, doi:10.3390/nu15112425.
  3. Luo, D.; Zhang, J.B.; Li, S.P.; Liu, W.; Yao, X.R.; Guo, H.; Jin, Z.L.; Jin, Y.X.; Yuan, B.; Jiang, H.; et al. Imperatorin Ameliorates the Aging-Associated Porcine Oocyte Meiotic Spindle Defects by Reducing Oxidative Stress and Protecting Mitochondrial Function. Frontiers in cell and developmental biology 2020, 8, 592433, doi:10.3389/fcell.2020.592433.

Point 13:

  • Line 218: what does urethane mean?

Respond 13:We appreciate the reviewer’s comments. I'm very sorry for the writing error. The word has been corrected.

Point 14:

  • Line 234: no values are provided for the results presented in Fig. 5E.

Respond 14:We appreciate the reviewer’s comments. The results of Fig. 5E have been supplemented based on the suggestions.

Point 15:

  • Line 265: What does mean “γH2A.X foci-positive cells in oocytes”? What cells are in an oocyte?

Respond 15:We appreciate the reviewer’s comments.γH2A.X is a marker representing DNA damage in oocytes.

The degree of DNA damage is calculated by the proportion of γH2A.X positive points to the total number of cells.

Oocyte: The oocyte that undergoes meiosis during oogenesis. It is divided into primary oocytes, secondary oocytes, and mature oocytes.

Point 16:

  • Lines 291-293: the sentence is not very clear. What do maturation-promoting factors (MPF) mean? Moreover, the Authors alternately use expression, level or activity, however, expression (especially the gene) is not always associated with its activity. This should be refined.

Respond 16:We appreciate the reviewer’s comments. MPF refers to the level of maturation promoting factor. We have reviewed the entire manuscript and made corrections to sentences that use the expression level or activity. Thank you for the reviewer's suggestions.

Point 17:

  • Lines 319 and 322: It would be worth adding information not only about the name of the cell line but also about the cell type.

Respond 17:We appreciate the reviewer’s comments. It has been added to the manuscript. Thank you for your suggestions.

Page 13, lines 331-336:

Previous research also demonstrated a dose-dependent reduction in ROS production in a human bronchial epithelial cell line (BEAS-2B) exposed to fine particulate matter, which could help alleviate respiratory diseases. Furthermore, eupatilin protected against hydrogen peroxide (H2O2)-induced oxidative stress and apoptosis by activating the PI3K/Akt signaling pathway in a human retinal pigment epithelium cell line (AR-PE-19)

Point 18:

  • Line 324: What does it mean that eupatilin can improve oxidative stress? Does this mean that it inhibits or stimulates this process?

Respond 18:We appreciate the reviewer’s comments. Through the results of this experiment, it was found that eupatilin can reduce the production of ROS in oocytes, increase the content of GSH, and reduce the levels of genes related to oxidative stress. Therefore, it is believed that eupatilin can reduce the occurrence of oxidative stress.

Point 19:

  • Lines 329-330: Please explain which genes are located on the endoplasmic reticulum?

Respond 19:We appreciate the reviewer’s comments. The endoplasmic reticulum is widely present in various eukaryotic cells and is a highly sensitive organelle. Any unfavorable factors may lead to disruption of the internal environment balance. Under normal circumstances, endoplasmic reticulum chaperone proteins, such as glucose regulating protein78 (GRP78), tightly bind to endoplasmic reticulum membrane proteins and become inactive, thereby maintaining the stability of endoplasmic reticulum. When endoplasmic reticulum stress occurs, a large number of unfolded and/or misfolded proteins aggregate in the endoplasmic reticulum cavity and competitively bind to GRP78, thereby activating unfolded protein response (UPR). In mammalian cells, UPR signaling is initiated by three types of endoplasmic reticulum transmembrane proteins, commonly referred to as the three branch sensors of UPR: activating transcription factor 6 (ATF6), protein kinase R-like endoplasmic reticulum kinase (PERK), and inositol requiring enzyme 1 (IRE1). Activating ATF6, PERK, and IRE1 will regulate the transcription of multiple downstream genes, thereby promoting correct protein folding and helping to rebuild ER homeostasis.

Reviewer 2 Report

Comments and Suggestions for Authors

The study aimed to investigate the functions of eupatilin in porcine oocyte maturation in vitro. This paper has reference value for how to improve the quality of oocyte maturation in vitro. However, I have some suggestions for the authors:

1. A representative picture of oocyte maturation should be presented in Figure 1.

2. SIRT1 is not a direct antioxidant enzyme, but rather a NAD+-dependent deacetylase, although it has been reported to mediate oxygen stress, and GSH, SOD2 may be more appropriate to evaluate eupatilin’ antioxidant capacity.

3. Some indicators of apoptosis, such as TUNLE, Bax and Caspase3, were detected in Figure 3, 4 and 6, which need to be shown in a figure.

4. In this study, how does the author evaluate maturation-promoting factors (MPF) activity, as there seems to be a lack of relevant evidence. The combination of CCNB1 and cdc2 forms MPF, and the detection of CCNB1 expression level alone is not helpful. More results should be needed to prove this, such as ELISA assays for MPF activity.

5. The authors suggest that exposure to eupatilin alleviates endoplasmic reticulum stress (ERS) in eggs, but that mRNA levels of ER-related genes alone are not well documented. More experiments need to be verified, such as using ER Tracker to evaluate ER quality.

Comments on the Quality of English Language

The language of the manuscript should be improved, for example line 229.

Author Response

Response to Reviewer 2 Comments

Comments and Suggestions for Authors

The study aimed to investigate the functions of eupatilin in porcine oocyte maturation in vitro. This paper has reference value for how to improve the quality of oocyte maturation in vitro. However, I have some suggestions for the authors:

Respond:We appreciate the reviewer’s comments. We have revised the manuscript based on your suggestion.

Point 1:

 A representative picture of oocyte maturation should be presented in Figure 1.

Respond 1:We appreciate the reviewer’s comments. We have made changes based on your suggestions.

Point 2:

SIRT1 is not a direct antioxidant enzyme, but rather a NAD+-dependent deacetylase, although it has been reported to mediate oxygen stress, and GSH, SOD2 may be more appropriate to evaluate eupatilin’ antioxidant capacity.

Respond 2:We appreciate the reviewer’s comments. Thank you for your suggestions so that we can make better revisions to the manuscript.

Point 3:

 Some indicators of apoptosis, such as TUNLE, Bax and Caspase3, were detected in Figure 3, 4 and 6, which need to be shown in a figure.

Respond 3:We appreciate the reviewer’s comments. TUNEL is obtained through immunofluorescence staining, while others are detected through real-time fluorescence quantitative PCR.

Point 4:

In this study, how does the author evaluate maturation-promoting factors (MPF) activity, as there seems to be a lack of relevant evidence. The combination of CCNB1 and cdc2 forms MPF, and the detection of CCNB1 expression level alone is not helpful. More results should be needed to prove this, such as ELISA assays for MPF activity.

Respond 4:We are sorry that due to the recent establishment of our laboratory, our experimental technology is not yet perfect, so there are some difficulties in conducting enzyme-linked immunosorbent assay experiments. Thank you for providing valuable suggestions. However, we still cannot give a definite answer to your question.

Point 5:

The authors suggest that exposure to eupatilin alleviates endoplasmic reticulum stress (ERS) in eggs, but that mRNA levels of ER-related genes alone are not well documented. More experiments need to be verified, such as using ER Tracker to evaluate ER quality.

Respond 5:We appreciate the reviewer’s comments. We plan to continue exploring the effects of eupatilin on the endoplasmic reticulum of oocytes in future experiments. Thank you very much for your suggestions.

Point 6:

Comments on the Quality of English Language

The language of the manuscript should be improved, for example line 229.

Respond 6:We appreciate the reviewer’s comments. We use the English editing website recommended by MDPI to edit articles in English. Thank you for your suggestion to make it better published in Animals magazine.

Round 2

Reviewer 1 Report

Comments and Suggestions for Authors

There are still inaccuracies in answering the question of what genes are located on the endoplasmic reticulum (as well as in the manuscript, lines: 395-397). Genes cannot be located on the ER. I think this is misconstrual. The ER contain proteins encoded by these genes. 

Line 326-327: it should be "γH2A.X focus-positive oocytes" not "γH2A.X focifocus-positive cells in oocytes". An oocyte is a single cell. Therefore, the statement "cells in oocytes" is incorrect.

I still think that all abbreviations used in the manuscript, figures and tables should be explained.

Author Response

Point 1:

There are still inaccuracies in answering the question of what genes are located on the endoplasmic reticulum (as well as in the manuscript, lines: 395-397). Genes cannot be located on the ER. I think this is misconstrual. The ER contain proteins encoded by these genes.

Respond 1:We appreciate the reviewer’s comments. Thank you sincerely for rectifying the issues we encountered, thereby facilitating our in-depth exploration of endoplasmic reticulum-related knowledge. We extend our heartfelt gratitude for your invaluable contribution to this article.

Point 2:

Line 326-327: it should be "γH2A.X focus-positive oocytes" not "γH2A.X focifocus-positive cells in oocytes". An oocyte is a single cell. Therefore, the statement "cells in oocytes" is incorrect.

Respond 2:We appreciate the reviewer’s comments. The article has been revised in accordance with your suggestions. We sincerely appreciate your valuable insights.

Point 3:

I still think that all abbreviations used in the manuscript, figures and tables should be explained.

Respond 3:We appreciate the reviewer’s comments. After a thorough review of the entire manuscript, we have provided detailed explanations for all abbreviations used in the text, figures, and tables.

Reviewer 2 Report

Comments and Suggestions for Authors

The authors had made well revisions, but there are still some minor ponits that need to be added before publication. For example, some abbreviation, such as  "COC","IVM", were explained for the first time , they are no longer necessary to re-interpret. 

Author Response

The authors had made well revisions, but there are still some minor ponits that need to be added before publication. For example, some abbreviation, such as  "COC","IVM", were explained for the first time , they are no longer necessary to re-interpret.

Respond:We appreciate the reviewer’s comments. Thank you sincerely for your valuable suggestions. We have incorporated your insightful recommendations into the manuscript, and we deeply appreciate your diligent efforts in refining our work.